# Human Adenovirus and Influenza A Virus Exacerbate SARS-CoV-2 Infection in Animal Models

**DOI:** 10.3390/microorganisms11010180

**Published:** 2023-01-11

**Authors:** Victor A. Svyatchenko, Vladimir A. Ternovoi, Roman Y. Lutkovskiy, Elena V. Protopopova, Andrei S. Gudymo, Nataliya V. Danilchenko, Ivan M. Susloparov, Nataliya P. Kolosova, Alexander B. Ryzhikov, Oleg S. Taranov, Vladimir V. Omigov, Elena V. Gavrilova, Alexander P. Agafonov, Rinat A. Maksyutov, Valery B. Loktev

**Affiliations:** State Research Center of Virology and Biotechnology “Vector”, 630559 Koltsovo, Novosibirsk Region, Russia

**Keywords:** SARS-CoV-2, influenza A virus, human adenovirus 5, coinfection, ferret, hamster

## Abstract

In this study, we investigated the features of the infectious process by simulating co-infection with SARS-CoV-2 and human adenovirus type 5 (HAdV-5) or influenza A virus (IAV) in vitro and in vivo. The determination of infectious activity of viruses and digital PCR demonstrated that during simultaneous and sequential HAdV-5 followed by SARS-CoV-2 infection in vitro and in vivo, the HAdV-5 infection does not interfere with replication of SARS-CoV-2. The hamsters co-infected and mono-infected with SARS-CoV-2 exhibited nearly identical viral titers and viral loads of SARS-CoV-2 in the lungs. The hamsters and ferrets co-infected by SARS-CoV-2- and IAV demonstrated more pronounced clinical manifestations than mono-infected animals. Additionally, the lung histological data illustrate that HAdV-5 or IAV and SARS-CoV-2 co-infection induces more severe pathological changes in the lungs than mono-infection. The expression of several genes specific to interferon and cytokine signaling pathways in the lungs of co-infected hamsters was more upregulated compared to single infected with SARS-CoV-2 animals. Thus, co-infection with HAdV-5 or IAV and SARS-CoV-2 leads to more severe pulmonary disease in animals.

## 1. Introduction

SARS-CoV-2 is a novel enveloped RNA-containing β-coronavirus that is phylogenetically related to SARS-CoV-1 and is the etiological agent that caused the COVID-19 pandemic in 2020–2021 [1,2]. Coinfection with SARS-CoV-2 and other viral respiratory infections may be an important factor in the developing COVID-19 pandemic. Coinfection with viral pathogens complicates both the diagnosis and the optimal choice of treatment. Moreover, it can modify the clinical manifestations and enhance disease severity, which can lead to increased mortality. It was shown that 3–21% of COVID-19 patients were also infected with other viral respiratory pathogens [3]. Approximately 5–7% of these coinfections are associated with human adenoviruses, which can cause adenovirus-related community acquired pneumonia (CAP) [4,5,6]. In the context of the coronavirus and annual influenza epidemics, the potential consequences of mixed infections by the novel coronavirus and influenza viruses are serious [7]. SARS-CoV-2 infects type II pneumocytes, which are also targets for influenza A virus (IAV) replication [8,9]. The coincidence of the COVID-19 pandemic and seasonal influenza outbreaks could put a large population at high risk of contracting these viruses simultaneously [10].

Small animal models are essential tools for viral pathogenesis, transmission, immunology and coinfection investigations [11,12,13,14,15]. The golden Syrian hamster and ferret models are usually used for SARS-CoV-2 and influenza infections. In SARS-CoV-2 challenge experiments, inoculated hamsters showed progressive weight loss with lethargy, ruffled fur, hunched back posture, and rapid breathing, with recovery by 14 days after virus inoculation [11]. The virus replicates to high titer in the upper and lower respiratory tracts, and the 50% infectious dose in Syrian hamsters is only five TCID_50_ [12]. SARS-CoV-2 causes pathological lung lesions including pulmonary edema and consolidation with evidence of interstitial pneumonia [12]. The Syrian hamster model is also the most commonly used human adenovirus animal model. Syrian hamsters that were used in HAdV-5 pathogenesis studies were found to have viral titers in blood and organ samples, and serologic as well as histologic evidence of infection [13]. Ferrets are commonly used in studies of influenza virus pathogenesis and transmission due to similarities to humans in receptor distribution and clinical course of the disease [14]. Ferrets are also susceptible to SARS-CoV-2 and the infected ferrets showed mild clinical signs including elevated body temperature and reduced activity, but no detectable body weight loss [15]. Viral shedding was mainly observed in the upper respiratory tracts but infectious viral titers were relatively low (1.83–2.88 lg10 TCID_50_/mL) [15].

The aim of this work was to study the features of the infectious process by simulating coinfection with SARS-CoV-2 and human adenovirus type 5 (HAdV-5, DNA-containing virus) or influenza A virus (IAV, RNA-containing virus) in vitro and in vivo using small animal models.

## 2. Materials and Methods

### 2.1. Viruses and Cell Cultures

In early 2020, the hCoV-19/Russia/StPetersburg-64304/2020 (GISAID, EPI_ISL_428868) SARS-CoV-2 strain was isolated from Vero E6 cells in Russia from a patient suffering from COVID-19. The adenovirus 5 strain “Adenoid 75” was obtained from ATCC (Manassas, VA, USA). Influenza A virus A/California/07/2009 (H1N1) pdm09 was obtained from the CDC (Manassas, VA, USA). Vero E6, HEK293A, and MDCK cells were obtained from the SRC VB “Vector” (Koltsovo, Novosibirsk Region, Russia) collection and were grown using DMEM (BioloT Ltd., Saint-Petersburg, Russia) in the presence of 10% fetal bovine serum (HyClone, Logan, UT, USA), penicillin (100 IU/mL), and streptomycin (100 μg/mL). Viral stocks of SARS-CoV-2, HAdV-5, and IAV with infectious titers of 6.5 lg TCID_50_/mL, 8.0 lg TCID_50_/mL, and 7.5 lg TCID_50_/mL, respectively, were stored at −70 °C.

### 2.2. Animals

Sixty-eight Syrian hamsters (*Mesocricetus auratus*) of both sexes and that were six-to-seven weeks old were obtained from the Centre for Genetic Resources of Laboratory Animals of the Centre Institute of Cytology and Genetics of the SB RAS and sixteen 0.7–1.3 kg ferrets (*Mustela putorius furo*) of both sexes from the SRC VB “Vector” laboratory animal nursery were used in the studies. The experimental animals were fed a standard diet and had access to water ad libitum, according to veterinary legislation and the requirements for humane animal care and the use of laboratory animals (National Research Council, 2011, Washington, DC, USA). The animal experiments were approved by the Bioethics Committee of the State Research Center of Virology and Biotechnology “Vector”.

### 2.3. Coinfection with HAdV-5 and SARS-CoV-2 In Vitro

The effects of coinfection in vitro were evaluated with simultaneous and sequential infections with HAdV-5 and SARS-CoV-2. Vero E6 cells were cultured in 24-well plates and inoculated in triplicate, either with both SARS-CoV-2 and HAdV-5 simultaneously (at a MOI of 0.1 TCID_50_ for each virus) or solely with HAdV-5 and 1 or 3 days later with SARS-CoV-2. Cells monoinfected with SARS-CoV-2 or HAdV-5 and mock-infected cells were used as controls. To determine the viral infectious titers and viral loads, the cells were collected at 24 h, 48 h, and 72 h postinfection (in the experiments with HAdV-5 preinfection, cells were collected at 24 h, 48 h, and 72 h after infection with SARS-CoV-2). The viral titers for SARS-CoV-2 and HAdV-5 in Vero E6 and HEK293A cells, respectively, were determined using a 50% tissue culture infectious dose (TCID_50_) assay as estimated by microscopic scoring of the CPE (Appendix A) and by measuring cell viability in the formazan-based MTT assay described previously [16,17].

### 2.4. Coinfection In Vivo

SARS-CoV-2 and HAdV-5 coinfection was studied in vivo using Syrian hamsters. The animals were randomly assigned to multiple groups (n = 9 animals per group): “CoV” (SARS-CoV-2)—on Day 0, the Syrian hamsters were intranasally challenged with SARS-CoV-2 (10^5^ TCID_50_); “CoV/ad” (SARS-CoV-2/HAdV5)—on Day 0, the Syrian hamsters were simultaneously intranasally challenged with SARS-CoV-2 (10^5^ TCID_50_) and HAdV-5 (10^6^ TCID_50_); “ad/3-days/CoV”—on Day 0, the hamsters were intranasally challenged with HAdV-5 (10^6^ TCID_50_), and, on Day 3, they were challenged with SARS-CoV-2 (10^5^ TCID_50_); “ad”(HAdV5)—on Day 0, the Syrian hamsters were intranasally challenged with HAdV-5 (10^6^ TCID_50_); “control”—mock-infected animals. Intranasal inoculation was performed by anaesthetizing the hamsters with isoflurane (Isothesia; Henry Schein Animal Health) and then inoculating the nostrils with the viruses in 150 µL of phosphate buffered saline (PBS). The control animals were administered PBS. After challenge, the hamsters were observed and weighed daily. On Day 4 postinfection (for the ad/3-days/CoV group, this took place on Day 4 after the challenge with SARS-CoV-2), the lungs were collected from three hamsters from each group. The lung tissues were homogenized and used to determine the infectious titers of the viruses and viral RNA/DNA loads. The SARS-CoV-2 and HAdV-5 titers, expressed as the 50% tissue culture infectious doses (TCID_50_), were determined by the cytopathic effect (CPE) assay in Vero E6 and HEK293A cells, respectively. The lung homogenates were analyzed for viral genome load by digital polymerase chain reaction (dPCR). The SARS-CoV-2 and HAdV-5 viral genome loads were determined using primers targeting the *1ab* and *hexon* genes, respectively. On Day 6 postinfection (for the group ad/3 days/CoV, this took place on Day 6 after the challenge with SARS-CoV-2), the lung tissues from three hamsters were fixed in formalin and used for the pathohistological study [18]. The slide samples were stained using a standard hematoxylin–eosin staining procedure. Light-optical examination and microphotography were carried out using an Imager Z1 microscope (Zeiss, Göttingen, Germany) equipped with a high-resolution HRc camera. The images were analyzed using the AxioVision Rel.4.8.2 software package (Carl Zeiss MicroImaging GmbH, Jena, Germany).

Coinfection with SARS-CoV-2 and IAV was studied in both ferret and hamster models. On Day 0, ferrets and hamsters were intranasally infected with 10^2^ TCID_50_ and 2 × 10^3^ TCID_50_ of IAV, respectively. Then, the ferret (on Day 2) and hamster (Day 4) groups were intranasally infected with 3 × 10^5^ TCID_50_ and 10^4^ TCID_50_ of SARS-CoV-2, respectively. After the challenge, the animals were observed and weighed daily. On Days 6, 8, and 10 after infection with IAV for the hamsters and on Days 4, 6, and 8 after infection with IAV for the ferrets, nasopharyngeal wash samples were collected from the animals to determine the viral genome loads. Animals were anaesthetized with intramuscular injection of tiletamine/zolazepam (Zoletil 100, Delpharm Tours, Chambray-lès-Tours, France) + xylazine (Xila, Interchemie, Venray, Estonia) at a dose of 30 + 10 mg/kg for hamsters and 5 + 3 mg/kg for ferrets; nasopharyngeal washes in ferrets and hamsters were obtained by administering 500 µL and 100 µL PBS to each nostril, respectively. On Day 10 post-monoinfection with IAV or on Day 10 postinfection with IAV and Day 6 postinfection with SARS-CoV-2 for the double-infected animals, the lungs were collected from the hamsters, and the lung tissues were fixed in formalin and used for the pathohistological study. Animal euthanasia was carried out using an automated compact CO_2_ system for humane output from the experiment of laboratory animals (Euthanizer, Moscow, Russia). The concentration of carbon dioxide (30% at the 1st stage, 70% at the 2nd stage) and gas supply rate satisfy the requirements of the American Veterinary Medical Association (AVMA) 2020.

### 2.5. RNA/DNA Extraction and qPCR

The viral loads of SARS-CoV-2 and IAV were quantified via quantitative PCR with reverse transcription (qRT–PCR). Viral RNA and DNA were extracted from the experimental samples (clarified cell lysates, lung homogenates, and nasopharyngeal washes) and purified with an AmpliPrime RIBO-prep Kit (K2-9-Et-100; Interlabservice, Moscow, Russia) in accordance with the manufacturer’s instructions. The purified RNAs were subjected to reverse transcription using a Reverta-L Kit (Interlabservice, Moscow, Russia).

For the quantification of SARS-CoV-2 and IAV, the Vector-PCRrv-2019-nCoV-RG Kit (SRC VB “Vector”, Koltsovo, Novosibirsk Region, Russia) and AmpliSens^®^ Influenza virus A/B-FL Kit (Moscow, Russia) reagent kits were used. The viral RNA loads were measured using primers targeting the SARS-CoV-2 1ab gene and IAV M gene (Appendix A). Thermal cycling was performed in a RotorGene 6000 (Bio–Rad, Hercules, CA, USA). Standard curves were generated via the 10-fold serial dilution of the Internal Positive Control Samples (IPCS) supplied with the respective PCR Kit (the SARS-CoV-2 and IAV reverse genetics plasmids encoding the 1ab and M genes, respectively) from 106 to 0.1 copies/reaction. The sample Ct values were obtained on two fluorescent channels, for viral cDNA and for IPCS. The viral cDNA Ct values were scaled relative to the IPCS Ct values.

Digital PCR (dPCR) was also used for the determination of the SARS-CoV-2 and HAdV-5 viral genome loads using the following primers (Appendix A) [14]. The reaction mixture contained ddPCRSupermix (×2) (Bio–Rad), primers (900 nM), a probe (250 nM), and cDNA or DNA. Each reaction mixture was converted into an oil-in-water emulsion using a QX200 droplet generator (Bio–Rad, Hercules, CA, USA). The resulting emulsion was transferred to a 96-well plate and incubated at 95 °C for 10 min to form microdroplets and then amplified in a C1000 Touch thermal cycler (Bio–Rad, Hercules, CA, USA) for 40 cycles with the following parameters: 95 °C for 10 min, 94 °C for 30 s, 56 °C for 15 s, 60 °C for 45 s, and then 98 °C for 10 min. The plate was transferred to a QX200 drop reader (Bio–Rad, Hercules, CA, USA), and the readout data were analyzed using QuantaSoft software (V1.7.4, Bio–Rad, Hercules, CA, USA).

The coefficient of variation for qPCR was calculated using the following formula: CVp, % = Ct (standard deviation)/Ct (mean value) × 100% (for five standard samples). The linear range of the dPCR was determined by estimating the average cDNA copy number in a microdroplet [19]. To assess the relative error, Poisson’s test was used.

To determine the chemokine and cytokine response, total RNA in lysed lung tissues was extracted with the RNeasy Mini kit (Qiagen, Hilden, Germany) and reverse-transcribed to cDNA with a TranscriptorFirst Strand cDNA Synthesis Kit (Roche, Basel, Switzerland). qRT–PCR using gene-specific primers [11] was performed as previously described [11].

### 2.6. Statistical Analysis

Basic statistical analyses, including calculations of the mean, standard deviation, and coefficient of variation of the mean Ct value, were performed using Excel (Microsoft Corp., Redmond, WA, USA). Statistical data processing was conducted using the statistical program STATISTICA 12 (StatSoft Inc., Tulsa, OK, USA).

Statistical evaluation of the differences between the groups was performed using the Student’s *t*-test; *p*  <  0.05 was considered significant.

### 2.7. Biosafety

All experiments involving any infectious viral materials were conducted in a Biosafety Level-3 Laboratory with all applicable national certificates and permissions required for studying SARS-CoV-2.

## 3. Results

### 3.1. Coinfection with SARS-CoV-2 and HAdV-5

Coinfection with SARS-CoV-2 and HAdV-5 in vitro was simulated by simultaneous and sequential infections of Vero E6 cells. We found that infection with HAdV-5 did not interfere with the replication of SARS-CoV-2 during the simultaneous coinfection of Vero E6 cells (Table 1A). The SARS-CoV-2 titers for monoinfection and coinfection were similar. The results of the dPCR for the viral genome loads correlated with the direct determination of infectious activity. A decrease in SARS-CoV-2 titers at 72 h postinfection may have been due to the inactivation of virions during the long incubation period with lysed cells. Lower HAdV-5 titer was noted in coinfected cells at 48 and 72 h postinfection due to SARS-CoV-2 induced lysis. Upon sequential infection of Vero E6 cells with HAdV-5 and 1 day later with the SARS-CoV-2 virus (Table 1B), the SARS-CoV-2 infectious titers during coinfection did not significantly differ from the titers during SARS-CoV-2 monoinfection.

The SARS-CoV-2 virus was able to effectively replicate in cells that had been preinfected with HAdV-5 3 days previously and already contained adenovirus at high titers, i.e., 4.7 lgTCID_50_/mL, at the time of infection with SARS-CoV-2. The coronavirus infectious titers within the HAdV-5-preinfected cells at 24 h and 48 h after infection with the SARS-CoV-2 virus did not significantly differ from the control values, while high concentrations of HAdV-5 were determined in cells for these periods (7.2 lgTCID_50_/mL).

The coinfection of Syrian hamsters with SARS-CoV-2 and HAdV-5 and monoinfection with SARS-CoV-2 resulted in clinical signs such as: lethargy, ruffled fur, hunched back posture, and rapid breathing. Subjectively, these manifestations were slightly more pronounced in the coinfected hamsters (the clinical scores [20] for “CoV/ad”, “ad/3-days/CoV”, and “CoV” groups were 3.60 ± 1.15, 3.85 ± 1.30, and 2.70 ± 0.80, respectively). In the same groups, a significant reduction in average animal weight was observed, i.e., 11% and 13%, respectively. Infection with SARS-CoV-2 alone resulted in a 9% reduction in weight (Figure 1). Animals that were intranasally monoinfected with HAdV-5 did not lose weight, but they did not gain body weight during the observation period either (their mean body weight values significantly differed both from the uninfected control group and from the SARS-CoV-2 and SARS-CoV-2/HAdV-5 groups; *p* < 0.05) (Figure 1). There were no visible clinical manifestations of infection in the group infected only with HAdV-5. These data demonstrate that when animals are coinfected with SARS-CoV-2 and HAdV-5, SARS-CoV-2 is the main contributor to the registered clinical manifestations of the infectious process. However, coinfection with HAdV-5 also led to an increase in the severity of the clinical manifestations.

To determine whether coinfection with SARS-CoV-2 and HAdV-5 was cooperative or competitive, the lung tissues collected from the infected hamsters were homogenized to determine the viral infectious titers and RNA/DNA genome loads (Figure 2A,B). Hamsters coinfected and monoinfected with SARS-CoV-2 exhibited nearly identical viral titers of SARS-CoV-2 (Figure 2A). Hamsters infected with HAdV-5 alone displayed significantly higher levels of infectious HAdV-5 in the lungs than hamsters coinfected with HAdV-5 and SARS-CoV-2 (lg TCID_50_/g; 6.9 vs. 4.9). The viral genome loads for both SARS-CoV-2 and HAdV-5 generally corresponded with the infectious viral titers (Figure 2B). Moreover, an approximately a 10-fold decrease in the SARS-CoV-2 genome loads was detected for hamsters preinfected with HAdV-5 and, 3 days later, infected with SARS-CoV-2 (7.3 vs. 8.5 lg genome copies/g; *p* < 0.05). Despite this, the infectious SARS-CoV-2 titers in the lungs of these animals did not differ significantly.

The HAdV-5 genome load did not significantly differ in the animals simultaneously infected with SARS-CoV-2 and HAdV-5 or for the animals infected only with HAdV-5. A decrease in the HAdV-5 viral genome loads in the lungs sequentially infected with HAdV-5 and SARS-CoV-2 was consistent with the results of the quantitative determination of infectious HAdV-5. Moreover, a significant decrease in the HAdV-5 titers in the lungs preinfected with adenovirus and, 3 days later, infected with SARS-CoV-2 was described previously for hamsters intranasally infected with the adenovirus on Day 6 postinfection [16].

### 3.2. Coinfection with SARS-CoV-2 and Influenza A Virus

The study of IAV and SARS-CoV-2 coinfection was carried out in hamster and ferret models. The effect of coinfection was assessed by determining the viral genome loads in the upper respiratory tracts of ferrets and hamsters coinfected with SARS-CoV-2 and IAV compared with those in monoinfected animals (Figure 3). Mixed infection with IAV preinfection in ferrets strongly (~4.0–5.0 lg) suppressed the replication of SARS-CoV-2 without alterations in IAV replication (Figure 3C,D). Mixed infection with IAV preinfection in hamsters slightly (~1.0 lg) reduced the SARS-CoV-2 viral genome loads on Day 2 but not on Days 4 and 6 post-SARS-CoV-2 challenge (Figure 3A). Mixed infection also slightly (~1.0 lg) increased the IAV viral load on Day 6 post-IAV infection (Figure 3B). Thus, the prior infection of ferrets with IAV critically inhibited the infectious activity of SARS-CoV-2. At the same time, previous infection with IAV in hamsters did not significantly affect the replication of SARS-CoV-2 in the respiratory tract. Notably, hamsters preinfected with IAV followed by secondary infection with SARS-CoV-2 exhibited a slight increase in IAV viral genome load in the respiratory tract on Day 6 after primary IAV inoculation compared to animals monoinfected with IAV.

When ferrets were infected with IAV, a reduction in the average animal weight was registered (7%, *p* < 0.05), whereas monoinfection with SARS-CoV-2 did not lead to significant differences in weight from the control group during the entire observation period (12 days). In the coinfected ferrets (infected with IAV and, 2 days later, with SARS-CoV-2), the average weight loss reached 11% (*p* < 0.05) (Figure 4A). The infection of hamsters with IAV was not accompanied by significant weight loss; in the group of hamsters infected with SARS-CoV-2, a 6% (*p* < 0.05) weight loss was registered. In the group of hamsters sequentially infected with IAV and, 4 days later, with SARS-CoV-2, a weight loss 4% (*p* < 0.05) greater than that of animals infected with SARS-CoV-2 alone was observed (Figure 4B). Thus, coinfection was accompanied by greater weight loss than monoinfection. These data, along with other manifestations (lethargy, ruffled fur, hunched back posture, and rapid breathing), indicate that mixed infection with SARS-CoV-2 and IAV in ferrets and hamsters increases the clinical severity.

### 3.3. Morphological Examination of Coinfected Animals

The morphological examination of the lungs of hamsters infected with HAdV-5 revealed moderately severe pathological manifestations characteristic of viral pneumonia (Figure 5B,F); i.e., there were single small areas of edema of the interalveolar septa (no more than a few percent of the section area); there was a moderate decrease in airiness according to the dystelectasis type (approximately 10–15% of the section area) in combination with an increase in blood flow in the interalveolar septa capillaries and a moderate infiltration of inflammatory cells of a mixed composition; there were no signs of vasculitis or bronchiolitis; the rest of the parenchyma exhibited essentially normal histological characteristics. In animals infected with SARS-CoV-2, the pathomorphological manifestations were significantly more pronounced, with signs of an acute phase of diffuse alveolar damage (Figure 5D); for example, there was dense edema in 40–50% of the section area with pronounced inflammatory cell infiltration, and there were distinct manifestations of vasculitis and bronchiolitis, with the rest of the parenchymal space being in a state of moderate emphysema. In animals simultaneously and sequentially coinfected with HAdV-5 and SARS-CoV-2 (Figure 5C,E,G–K), large foci of dense edema and pronounced polymorphic (mainly lymphocytic) infiltration were revealed, occupying up to 70% of the total sectional area. The infiltrate also contained neutrophils, plasma cells, and histiocytes. A considerable number of large cells with fine chromatin and, occasionally, with nucleolus were present in groups or individually. There were also pronounced signs of vasculitis and bronchiolitis, large foci of plasma, and hemorrhages (Appendix A). Thus, the lung histological data illustrate that HAdV-5 and SARS-CoV-2 coinfection induces more severe pathological changes in the lungs than HAdV-5 or SARS-CoV-2 monoinfection.

The obtained data indicate that adenovirus does not reduce the replication efficiency of SARS-CoV-2 in the lungs and that coinfection with SARS-CoV-2 and adenovirus increases the severity of the pathological manifestations.

A histopathological study of IAV and SARS-CoV-2 coinfection was carried out in hamsters. Epithelial cell degeneration (Figure 5L), vasculitis (Figure 5M), and perivascular lymphocyte infiltration (Figure 5N) were observed in the lungs of IAV-infected animals. Monoinfection with SARS-CoV-2 6 days after intranasal inoculation was associated with histological changes identical to those described above. SARS-CoV-2- and IAV-associated changes were also observed in the coinfected hamsters (at 10 days and 6 days after IAV and SARS-CoV-2 infection, respectively), and they were generally more pronounced with dense consolidation of lung tissue caused by edema and massive cell death, mainly caused by lymphocytic infiltration and alveolar necrosis (Figure 6D,E; Appendix A). Thus, IAV and SARS-CoV-2 coinfection induced more severe lung pathological changes than IAV or SARS-CoV-2 monoinfection.

Staining with hematoxylin and eosin. All images were taken using a 40× lens. The scale bar is shown in the pictures.

### 3.4. Chemokine/Cytokine Responses in Coinfected Animals

SARS-CoV-2 infection resulted in increased mRNA levels of genes involved in the interferon and cytokine signaling pathways. The cytokine/chemokine profile in the lungs of SARS-CoV-2-infected hamsters exhibited a time-dependent trend of gene expression (Figure 7). Interferon gamma mRNA level peaked by day 4–6 post infection, implying that SARS-CoV-2 triggered the innate immune response. Proinflammatory chemokine/cytokine mRNA levels peaked at 6 days post infection, which represented the activation of inflammation and virus-induced cell death. At 12 days post infection, the cytokine/chemokine profile dropped to the basal level and indicated the resolution of acute inflammation. When hamsters were infected with both SARS-CoV-2 and IAV or with HAdV-5, the mRNA levels of certain genes within these pathways remained increased in abundance at later time points in comparison to single SARS-CoV-2 infection (Figure 7); these included interferon gamma and proinflammatory cytokines such as interleukin-6, CCL17, and TGF-β. Dysregulated interferon and cytokine responses in coinfected animals can result in enhanced lung tissue damage.

## 4. Discussion

Acute infections of the lower respiratory tract are one of the leading causes of disability and death worldwide [21]. Many respiratory viruses can cause community-acquired viral pneumonia, e.g., respiratory syncytial virus, influenza viruses, rhino-/enteroviruses, parainfluenza viruses, adenoviruses, and human metapneumovirus. The simultaneous circulation of SARS-CoV-2 and seasonal respiratory viruses has the potential to cause serious public health problems.

Influenza viruses are one of the main etiological causes of viral pneumonia. Recently, several published studies that focused on coinfection with IAV and SARS-CoV-2 in humans reported severe outcomes and a higher risk of death [22,23,24]. In contrast, other studies reported mild symptoms in coinfected patients [25]. HAdV is important in the development of CAP in both immunocompetent and immunocompromised individuals [26]. Adenovirus-related CAP was ranked in the top 10 most common causes of CAP [27,28]. HAdV is easily transmittable and thus highly contagious [29]; for this reason, mixed infection with SARS-CoV-2 is very likely.

The clinical manifestations of SARS-CoV-2 and seasonal respiratory infections are fairly similar, including cough, fever, and pneumonia [3]. These infections are usually airborne, which allows pathogens to infect nasal, bronchial, and alveolar epithelial cells [30,31]. Moreover, alveolar type II cells are some of the most important target cells for SARS-CoV-2, IAV, and other seasonal respiratory viral infections [9]. Influenza viruses, SARS-CoV-2, and other respiratory viruses are characterized by increased permeability of the alveolar capillary membrane, which leads to alveolar overflow and respiratory failure. This is known as acute respiratory distress syndrome [32,33]. This makes an increase in the severity of the disease with coinfection by these viruses a real possibility.

The results presented in our work demonstrate that during the simultaneous and sequential (DNA and RNA viruses) infection of cells in vitro, HAdV-5 infection did not interfere with SARS-CoV-2 replication. Of particular importance is the fact that SARS-CoV-2 replicates in preinfected cells as well as those already containing HAdV-5 at high titers with the same efficiency as in intact cells. Our in vivo study demonstrated that preinfection with a DNA virus (HAdV-5) did not decrease the infectivity of an RNA virus (SARS-CoV-2). However, the coinfected hamsters displayed more pronounced lung damage and clinical manifestations than the SARS-CoV-2- or HAdV-5 monoinfected animals.

These results demonstrate the absence of strong viral interference (competitive suppression) between HAdV-5 and SARS-CoV-2 in vitro and in vivo. Aside from viral interference, i.e., when one virus inhibits the replication of another coinfecting virus, coinfections with certain viruses may also induce an increase in viral replication, although coinfections sometimes have no effect on virus replication.

In a previous study, an enhancement of SARS-CoV-2 replication was detected after preliminary IAV infection of cell cultures and K18-hACE2 mice [34]. This enhancement was associated with increased expression of ACE2, which is a major receptor for SARS-CoV-2 entry into host cells. A 2- to 3-fold increase in ACE2 mRNA levels was detected after IAV infection of A549 cells. However, a 28-fold increase in ACE2 mRNA levels was detected after IAV and SARS-CoV-2 coinfection. Moreover, preinfection with IAV increased the SARS-CoV-2 genome load for A549 cells (by nearly 10–15-fold). The authors suspected that IAV infection mildly induced the expression of ACE2, which encourages SARS-CoV-2 entry into cells. Additionally, a significant increase in the SARS-CoV-2 viral genome load was observed in the lungs of coinfected mice (a 6.6–12.9-fold increase). The lung histological data also illustrate that IAV and SARS-CoV-2 coinfection induced more severe lung pathological changes—with massive cell infiltration and obvious alveolar necrosis—than SARS-CoV-2 monoinfection [34].

Alternatively, as has recently been reported, K18-hACE2 transgenic mice infected with SARS-CoV-2 alone displayed significantly higher levels of SARS-CoV-2 RNA at Day 3 postinfection than mice preinfected with IAV [22]. Thereafter, on Day 7 after SARS-CoV-2 inoculation, the coinfected and monoinfected mice exhibited nearly identical levels of SARS-CoV-2 RNA. In this study, sequential infection with IAV followed by SARS-CoV-2 led to more severe pulmonary disease than infections with IAV or SARS-CoV-2 alone [22].

Through experimental coinfections with IAV and SARS-CoV-2, we found that IAV preinfection in hamsters caused an almost 10-fold decrease in the levels of SARS-CoV-2 RNA in the respiratory tracts on Day 2 after SARS-CoV-2 inoculation compared to those in monoinfected animals. However, on Days 4 and 6, the coinfected and monoinfected hamsters exhibited similar levels of SARS-CoV-2 RNA. Interestingly, coinfected hamsters exhibited significantly lower levels of SARS-CoV-2 viral RNA at Day 2 than SARS-CoV-2 singly infected hamsters, indicating that while coinfection results in enhanced pulmonary damage, existing IAV infection interferes with SARS-CoV-2 infection. Our data are in accordance with those from a previous study using transgenic mice as a model [22]. In our study, the sequential infection of hamsters with IAV followed by SARS-CoV-2 led to more severe disease than infections with IAV or SARS-CoV-2 alone.

It was shown that coinfection with IAV causes more severe body weight loss and more severe and prolonged pneumonia in SARS-CoV-2-infected hamsters [35]. Each virus can efficiently spread in the lungs without interference by the other. However, in immunohistochemical analyses, SARS-CoV-2 and IAV were not detected at the same sites in the respiratory organs of co-infected hamsters. Unlike our work, they infected hamsters with IAV and SARS-CoV-2 at the same time and used a higher dose of SARS-CoV-2.

A qPCR analysis revealed that the expression of several genes specific to interferon and cytokine signaling pathways in the lungs of hamsters sequentially infected with IAV or HAdV-5 followed by SARS-CoV-2 was more upregulated than that in animals infected with SARS-CoV-2 alone. The interferon and cytokine responses (IL-6, CCL17 and TGF-β). in coinfected animals remained increased at later time points in comparison to those of monoinfected animals. A significant increase in IL-6 was detected in the sera of hamsters co-infected with SARS-CoV-2 and IAV, suggesting that IL-6 may be involved in the increased severity of pneumonia [35]. Dysregulated interferon and cytokine responses in coinfected animals can result in enhanced lung tissue damage.

Unlike the data described above, IAV preinfection of ferrets dramatically decreased the levels of SARS-CoV-2 RNA in the respiratory tract (approximately 4.0–5.0 lg) compared to those in monoinfected animals. As has recently been reported [36], IAV was detectable from all coinfected ferrets from nasal washes collected at 1, 3, and 5 days post infection. However, the SARS-CoV-2 was only detectable at 1 day postinfection in the coinfected ferrets while the virus was detectable for up to 5 days postinfection in SARS-CoV-2 single-infected ferrets. Moreover, no infectious SARS-CoV-2 was detected in the ferret lungs regardless of whether they were singly infected with SARS-CoV-2 or coinfected with IAV, indicating that SARS-CoV-2 infected or replicated efficiently in the upper but not the lower respiratory tract in ferrets. The presented results demonstrate a pronounced dependence of the replication efficiency of viruses during coinfection on the sensitivity of the laboratory model to a particular virus (ferrets are the most widely used animal model in influenza studies [37], while the Syrian golden hamster is considered the most adequate small animal model for researching the new coronavirus [13]). Our data and the data presented in [36] demonstrate that despite the strong inhibition of SARS-CoV-2 replication in ferrets infected with IAV, coinfected animals demonstrated more pronounced clinical manifestations than monoinfected animals.

Thus, coinfection with HAdV-5 or IAV and SARS-CoV-2 leads to more severe pulmonary disease in animals. However, any animal model is incapable of adequately demonstrating the consequences of coinfection in humans. Nevertheless, a large clinical SARS-CoV-2 co-infection study (227 patients had influenza viruses and 136 patients had adenoviruses) showed that SARS-CoV-2 co-infections with influenza viruses and adenoviruses were each significantly associated with increased odds of death [38]. The data presented here indicate that the prevention of both influenza and seasonal common cold infections (through vaccinations, the stimulation of innate immune responses, social distancing, respirator wearing, etc.) during the COVID-19 pandemic is of great importance. In severe SARS-CoV-2 cases during the influenza or adenovirus infection season, it is necessary to conduct additional diagnostic analyses and, if necessary, take them into account when selecting the treatment regimen.

## 5. Conclusions

Outbreaks of seasonal respiratory infections during the COVID-19 pandemic pose a potentially severe threat to public health. In this study, we investigated coinfection with human adenovirus 5, influenza A virus, and severe acute respiratory syndrome coronavirus 2 in vitro and in vivo. The results demonstrate that during simultaneous and sequential HAdV-5 infection followed by SARS-CoV-2 infection in vitro and in vivo, HAdV-5 infection does not inhibit SARS-CoV-2 replication. IAV preinfection moderately decreased the levels of SARS-CoV-2 replication in the respiratory tract of coinfected animals compared to that of monoinfected hamsters. However, coinfected HAdV-5 or IAV and SARS-CoV-2 hamsters displayed more pronounced lung damage and clinical manifestations than monoinfected animals. Thus, coinfection with HAdV-5 or IAV and SARS-CoV-2 leads to more severe pulmonary disease in animals. In conclusion, our study indicates that both influenza and seasonal common cold infections prevention during the SARS-CoV-2 pandemic are of great importance.

## Figures and Tables

**Figure 1 microorganisms-11-00180-f001:**
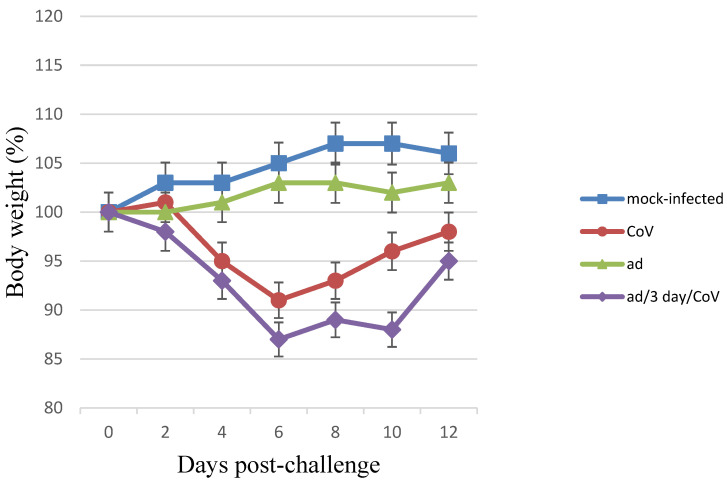
Body weight changes in monoinfected and coinfected with HAdV-5 and SARS-CoV-2 Syrian hamsters. Body weight changes in monoinfected and coinfected with HAdV-5 and SARS-CoV-2 Syrian hamsters. “CoV (SARS-CoV-2)”—on Day 0, the Syrian hamsters were intranasally challenged with SARS-CoV-2 (10^5^ TCID_50_); “ad (HAdV-5)/3-days/CoV”—on Day 0, the preinfected (3 days earlier) with HAdV-5 (10^6^ TCID_50_) hamsters were intranasally challenged with SARS-CoV-2 (10^5^ TCID_50_); “ad”—on Day 0, the Syrian hamsters were intranasally challenged with HAdV-5 (10^6^ TCID_50_); “mock-infected”—on Day 0, the hamsters were intranasally inoculated wits PBS. n = 9 at 0 dpi to 4 dpi; n = 6 at 5 dpi to 6 dpi as 3 animals were sacrificed; n = 3 at 7 dpi to 12 dpi as 3 animals were sacrificed. The values represent the means ± SDs of individual animals. Student’s *t*-test was used for two-group comparisons, *p* < 0.05.

**Figure 2 microorganisms-11-00180-f002:**
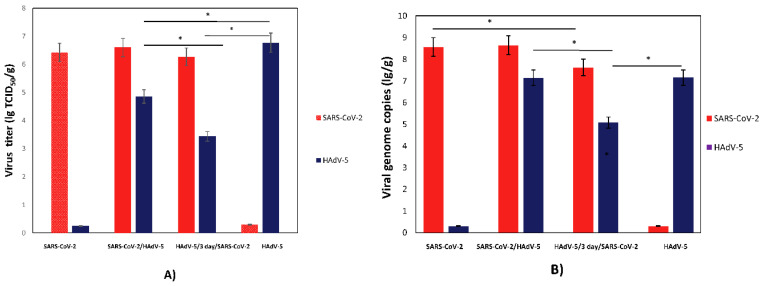
SARS-CoV-2 and HadV-5 replication in monoinfected and coinfected hamsters. (**A**) SARS-CoV-2 and HadV-5 viral infectious titers; (**B**) SARS-CoV-2 and HadV-5 viral genome loads. SARS-CoV-2—Syrian hamsters were intranasally challenged with SARS-CoV-2 (10^5^ TCID_50_); SARS-CoV-2/HadV-5—Syrian hamsters were simultaneously intranasally challenged with SARS-CoV-2 (10^5^ TCID_50_) and HadV-5 (10^6^ TCID_50_); HadV-5/3-days/SARS-CoV-2—Syrian hamsters were intranasally challenged with HadV-5 (10^6^ TCID_50_) and, 3 days later, with SARS-CoV-2 (10^5^ TCID_50_); ad—Syrian hamsters were intranasally challenged with HadV-5 (10^6^ TCID_50_). On Day 3 postinfection (for the ad/3 days/CoV group, this took place on Day 3 after the challenge with SARS-CoV-2), half of the lung tissues collected from hamsters were homogenized to determine the viral infectious titers and RNA/DNA genome loads. (A) The SARS-CoV-2 and HadV-5 titers, expressed as the 50% tissue culture infectious doses per gram (TCID_50_/g), were determined by the CPE assay in Vero E6 (Appendix A) and HEK293A (Appendix A) cells, respectively. (B) The lung homogenates were analyzed for viral genome loads by dPCR. The SARS-CoV-2 and HadV-5 genome loads were determined using primers targeting the *1ab* and *hexon* genes, respectively. The values represent the means ± SDs of three hamsters. Student’s *t*-test was used for two-group comparisons, * *p*< 0.05.

**Figure 3 microorganisms-11-00180-f003:**
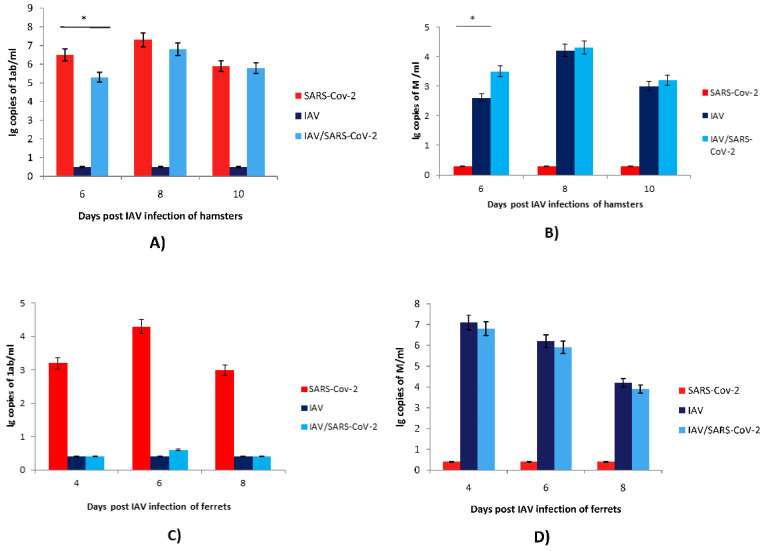
SARS-CoV-2 and IAV replication in monoinfected and coinfected hamsters and ferrets. (**A**,**C**) SARS-CoV-2 viral genome loads were determined using qPCR for the *1ab* gene in nasopharyngeal washes of hamsters and ferrets, respectively. (**B**,**D**) IAV viral genome loads were determined using qPCR for the *M* gene in nasopharyngeal washes of hamsters and ferrets, respectively. SARS-CoV-2—the Syrian hamsters and ferrets were intranasally challenged with SARS-CoV-2 (10^4^ TCID_50_ and 3 × 10^5^ TCID_50_, respectively); IAV—the Syrian hamsters and ferrets were intranasally challenged with IAV (2 × 10^3^ TCID_50_ and 10^2^ TCID_50_, respectively); IAV/SARS-CoV-2—the Syrian hamsters were intranasally challenged with IAV (2 × 10^3^ TCID_50_) and, 4 days later, with SARS-CoV-2 (10^4^ TCID_50_); the ferrets were intranasally challenged with IAV (10^2^ TCID_50_) and, 2 days later, with SARS-CoV-2 (3 × 10^5^ TCID_50_). On Days 6, 8, and 10 after infection with IAV for the hamsters and on Days 4, 6, and 8 after infection with IAV for the ferrets, nasopharyngeal washes were collected from all animals to determine the viral genome loads. The nasopharyngeal washes were analyzed by qPCR. The SARS-CoV-2 and IAV viral genome loads were determined using primers targeting the *1ab* and *M* genes, respectively. The values represent the means ± SDs of three individual animals. Student’s *t*-test was used for two-group comparisons, * *p* < 0.05.

**Figure 4 microorganisms-11-00180-f004:**
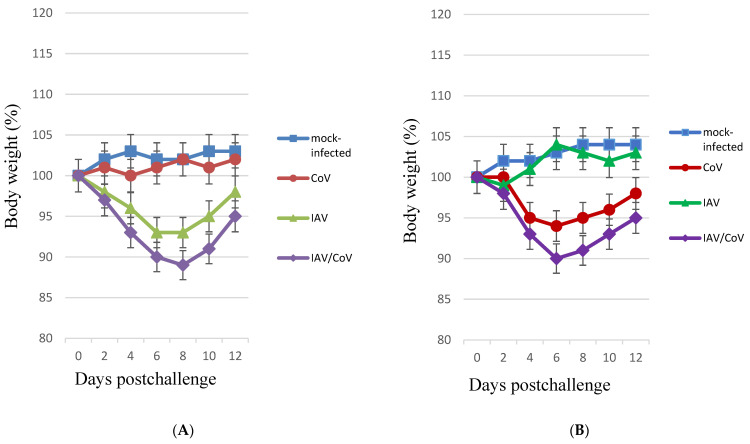
Body weight changes in monoinfected and coinfected with IAV and SARS-CoV-2 animals. (**A**) Body weight changes in monoinfected and coinfected with IAV and SARS-CoV-2 ferrets. “CoV (SARS-CoV-2)”—on Day 0, the ferrets were intranasally challenged with SARS-CoV-2 (3 × 10^5^ TCID_50_); “IAV”—on Day 0, the ferrets were intranasally challenged with IAV (10^2^ TCID_50_); “IAV/CoV”—on Day 0, the ferrets preinfected (2 days earlier) with IAV (10^2^ TCID_50_) were intranasally challenged with SARS-CoV-2 (3 × 10^5^ TCID_50_); “mock-infected”—on Day 0, the ferrets were intranasally inoculated with PBS. n = 4. (**B**) Body weight changes in monoinfected and coinfected with IAV and SARS-CoV-2 hamsters. “CoV SARS-CoV-2)”—on Day 0, the hamsters were intranasally challenged with SARS-CoV-2 (10^4^ TCID_50_); “IAV”—on Day 0, the hamsters were intranasally challenged with IAV (2 × 10^3^ TCID_50_); “IAV/CoV”—on Day 0, the hamsters preinfected (4 days earlier) with IAV (2 × 10^3^ TCID_50_) were intranasally challenged with SARS-CoV-2 (10^4^ TCID_50_); “mock-infected”—on Day 0, the hamsters were intranasally inoculated with PBS. n = 8 at 0 dpi to 6 dpi; n = 4 at 7 dpi to 12 dpi, as 4 animals were sacrificed. The values represent the means ± SDs of individual animals. Student’s *t*-test was used for two-group comparisons. * *p* < 0.05.

**Figure 5 microorganisms-11-00180-f005:**
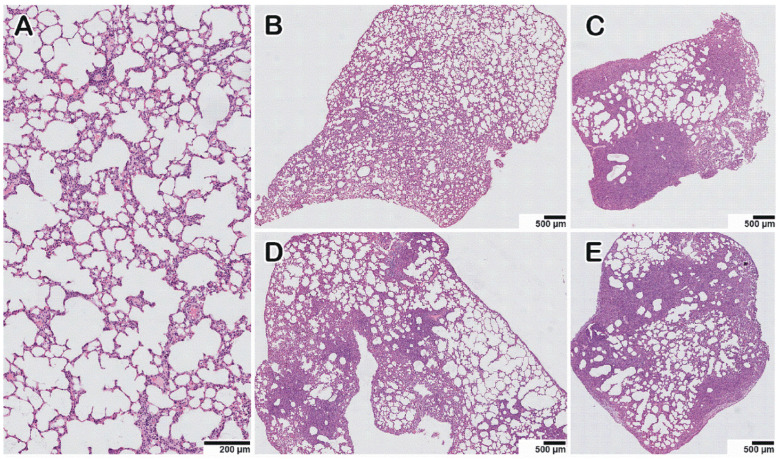
Histopathological alterations in hamster lungs monoinfected (HAdV-5, SARS-CoV-2, IAV) and coinfected with HAdV-5 and SARS-CoV-2. (**A**) mock-infected control animal, normal lung. (**B**–**E**) Representative images showing pathological changes in lung tissue after infection: (**B**)—with HAdV-5; (**C**)—with HAdV-5 and SARS-CoV-2 simultaneously; (**D**)—with SARS-CoV-2; (**E**)—with HAdV-5 and, 3 days later, with SARS-CoV-2. In both cases of coinfection with two viruses (**C**,**E**), there was a significant increase in the consolidation zones and pronounced perifocal compensatory atelectasis. (**F**) Manifestations of adenovirus infection: edema and mixed inflammatory cell infiltration of the interalveolar septa and, in the upper part of the image, pronounced plethora of the capillaries. (**G**–**K**) Typical pathological changes found in mixed infections. (**G**) Blood (black arrows), macrophages, and syncytium (white arrow) in the lumen of the alveoli and the activation and hyperplasia of type II pneumocytes. (**H**) Hemorrhage in the alveoli (black arrows). (**I**) Spasm of small vessels of the arterial type (black arrow); perivascular edema and perivascular lymphocytic infiltration (white arrow); desquamation of the epithelial lining of the bronchi (red arrow). (**J**) Destruction of the small bronchial wall (bronchiolitis, black arrow); desquamated bronchial epithelium in the lumen (red arrow); peribronchial and perivascular infiltration (white arrow). (**K**) Zone of consolidation of lung tissue with spasmodic arterioles (black arrow). (**L**–**N**) Manifestations of IAV infection: epithelial cell degeneration, vasculitis, and perivascular lymphocyte infiltration, respectively (black arrows).

**Figure 6 microorganisms-11-00180-f006:**
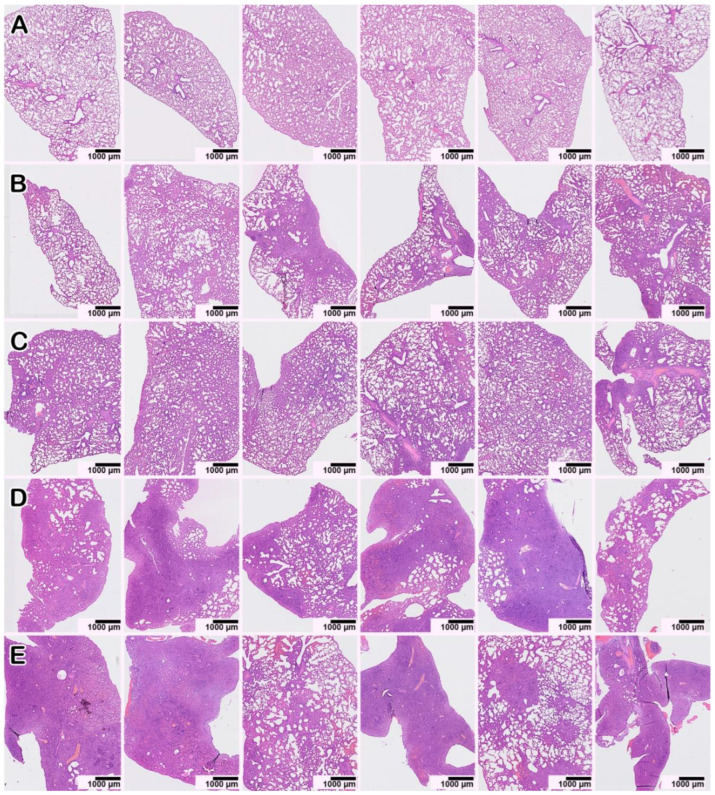
Pathological lesions in hamster lungs 10 days after monoinfection with IAV (**B**,**C**) or on Day 10 after infection with IAV and Day 6 after infection with SARS-CoV-2 for the case of coinfection (**D**,**E**). Each of the six survey images shows the lung tissue of one animal (the three on the left—the left lung; the three on the right—the right lung) to give a representative idea of the nature and prevalence of the pathology. (**A**) Mock-infected animal. (**D**,**E**) In double-infected animals, zones of dense consolidation of lung tissue caused by edema and cellular infiltration, mainly lymphocytic, spread to the entire anatomical lobe. Staining with hematoxylin–eosin. All images were taken using a 20× objective and are presented at the same scale. The scale bar is shown in the pictures.

**Figure 7 microorganisms-11-00180-f007:**
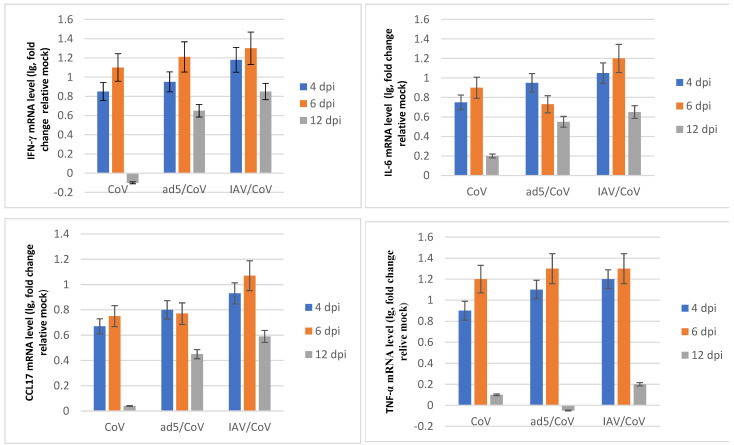
The chemokine/cytokine profile in the lungs of hamsters coinfected with SARS-CoV-2 and HAdV-5 or IAV. “CoV (SARS-CoV-2)”—on Day 0, the hamsters were intranasally challenged with SARS-CoV-2 (10^5^ TCID_50_); “ad5 (HAdV-5)/CoV”—on Day 0, the hamsters preinfected (3 days earlier) with HAdV-5 (10^6^ TCID_50_) were intranasally challenged with SARS-CoV-2 (10^5^ TCID_50_); IAV/CoV—on Day 0, the hamsters preinfected (4 days earlier) with IAV (2 × 10^3^ TCID_50_) were intranasally challenged with SARS-CoV-2 (10^4^ TCID_50_); dpi—day postinfection with SARS-CoV-2. The data are expressed as fold changes relative to the mock-infected group. The values represent the means ± SDs of three individual animals. Student’s *t*-test was used for two-group comparisons, *p* < 0.05.

**Table 1 microorganisms-11-00180-t001:** Modelling of HAdV-5 and SARS-CoV-2 coinfection in vitro during simultaneous (**A**) and consecutive (**B**) infection of Vero E6 cells (MOI 0.1 TCID_50_).

(A)						
	Monoinfection				Coinfection	
	SARS-CoV-2		HAdV-5		SARS-CoV-2 + HAdV-5	
Time postinfection	Viral titers, lgTCID_50_/mL	Viral RNA load according to dPCR, lg genome copies/mL	Viral titers, lgTCID_50_/mL	Viral DNA load according to dPCR, lg genome copies/mL	Viral titers, SARS-CoV-2/HAdV-5, lgTCID_50_/mL	Viral RNA/DNA load according to dPCR, SARS-CoV-2/HAdV-5, lg genome copies/mL
24 h	4.7 ± 0.3	7.6 ± 0.4	<1.0	<2.0	4.9 ± 0.3 ^n^/<1.0 ^n^	7.9 ± 0.3 ^n^/<2.0 ^n^
48 h	6.8 ± 0.4	8.8 ± 0.3	3.8 ± 0.3	6.3 ± 0.2	6.7 ± 0.4 ^n^/2.4 ± 0.3 *	8.2 ± 0.3 ^n^/3.8 ± 0.3 *
72 h	5.6 ± 0.3	6.9 ± 0.2	4.8 ± 0.4	6.9 ± 0.3	5.3 ± 0.3 ^n^/2.2 ± 0.2 *	6.5 ± 0.3 ^n^/3.7 ± 0.2 *
**(B)**						
**Time** **Postinfection with SARS-CoV-2**	**Viral titers (SARS-CoV-2/HAdV-5), lgTCID_50_/mL**		**Viral RNA/DNA load according to dPCR (SARS-CoV-2/HAdV-5), lg genome copies/mL**		**Viral titers (SARS-CoV-2), lgTCID_50_/mL**	**Viral RNA load by dPCR (SARS-CoV-2), lg genome copies/mL**
	HAdV-5 preinfection-24 h-SARS-CoV-2 infection				Mock preinfection-24 h-SARS-CoV-2 infection	
24 h	5.0 ± 0.3 ^n^/2.9 ± 0.2		7.9 ± 0.3 ^n^/5.1 ± 0.2		5.2 ± 0.4	7.7 ± 0.3
48 h	7.1 ± 0.4 ^n^/3.9 ± 0.4		8.9 ± 0.4 ^n^/6.1 ± 0.3		7.0 ± 0.4	8.7 ± 0.4
72 h	5.5 ± 0.3 ^n^/4.7 ± 0.3		7.4 ± 0.3 ^n^/6.2 ± 0.2		5.7 ± 0.3	6.7 ± 0.2
	HAdV-5 preinfection-72 h-SARS-CoV-2 infection				Mock preinfection-72 h-SARS-CoV-2 infection	
24 h	5.1 ± 0.3 ^n^/7.2 ± 0.4		7.6 ± 0.2 ^n^/7.9 ± 0.3		5.3 ± 0.3	7.3 ± 0.3
48 h	6.7 ± 0,4 ^n^/7.2 ± 0,4		8.6 ± 0.3 ^n^/8.4 ± 0.4		6.9 ± 0.4	8.8 ± 0.3
72 h	5.2 ± 0.3 ^n^/7.4 ± 0.3		7.2 ± 0.2 ^n^/8.1 ± 0.3		5.3 ± 0.3	7.1 ± 0.2

Values represent means ± SD of three independent experiments. Student’s *t*-test was used for two-group comparisons. * *p* < 0.05, ^n^—not statistically.

## Data Availability

The data presented in this study are available in the article.

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
