# Peer review of "Human Adenovirus and Influenza A Virus Exacerbate SARS-CoV-2 Infection in Animal Models"

_microorganisms, 2023, doi:10.3390/microorganisms11010180_

Round 1
Reviewer 1 Report
The manuscript "SARS-CoV-2 infection enhances by human adenovirus or influenza A virus in animal models" is a well-written paper devoted to the study of the features of the infectious process caused by coinfection of SARS-CoV-2 and human adenovirus or influenza A virus in vitro and in vivo. It is very interesting topic chosen by authors. The study shows that both influenza and HAdV5 cause more severe diseases when co-infected with SARS-CoV-2. The authors obtained really interesting results. I believe that this manuscript can be published in the Microorganisms.
I do have some comments on the text.
1. References in the text occur in the form of numbers [1] and (Bai et al., 2021).
2. There are no labels A, B, etc. in the figures. It is unclear where the signature to the figure ends and the text begins.
3. It is necessary to add to the discussion how the results of the study relate to clinical data. For example, in the article in Lancet (Swets et al., 2022) it is written: “SARS-CoV-2 co-infections with influenza viruses and adenoviruses were each significantly associated with increased odds of death”.
Author Response
Comments for Reviewer 1
Thank you for considering our manuscript for publication in Microorganisms for special issue "Advances in SARS-CoV-2 Infection" titled “SARS-CoV-2 infection enhances by human adenovirus or influenza A virus in animal models”
We thank you for valuable suggestions that allowed us to make the manuscript more convincing and understandable. We accepted your suggestion and made corresponding changes in the manuscript.
Below please find our detailed responses to your questions and comments. All changes into MS (figures) are marked by in yellow (see attachment)
Yours sincerely,
Merry Christmas and Happy New Years!!!
Prof. (Dr.) Valery B. Loktev
You wrote:
- References in the text occur in the form of numbers [1] and (Bai et al., 2021).
The MS has been revised and modified.
- There are no labels A, B, etc. in the figures. It is unclear where the signature to the figure ends and the text begins.
The MS (figures) has been revised and modified.
- It is necessary to add to the discussion how the results of the study relate to clinical data. For example, in the article in Lancet (Swets et al., 2022) it is written: “SARS-CoV-2 co-infections with influenza viruses and adenoviruses were each significantly associated with increased odds of death
The MS has been revised and modified.

Reviewer 2 Report
Thanks for letting me review this draft. Here are some suggestions (major and minor) that should be addressed during preparation of a revised manuscript.
11. Alternative title (suggestion): Human Adenovirus and Influenza A Virus Exacerbate SARS-CoV-2 Infection in Animal Models.
22. Line 7: Is “World-Class Genomic Research Center” really part of your official affiliation?
33. In the very short introduction paragraph, include more information on animal models of the viruses used, see doi: 10.1016/j.coviro.2021.03.009 (SARS-CoV-2), doi: 10.3390/biology10121253 (HAdV), and doi: 10.1002/2211-5463.13416 (IAV) for example. Focus on significant information on hamsters and ferrets!
44. In your M&M section, please indicate how many animals you used in total and how they were housed. Also, you mention the infection procedure incl. medicals used, but not how these animals were euthanized. That should be included.
55. Line 85 and elsewhere: use superscript numbers (105 instead of 105 TCID50)
66. Were the nasopharyngeal wash samples collected under general anesthesia! Please clarify and include this information in the M&Ms if appropriate.
77. Line 137f: You used dPCR to compare genome loads between HAdV and SARS-CoV-2… a DNA virus vs. an RNA virus. For both, it seems you used cDNA as PCR sample, correct (see line 140)? Please clarify and revise.
88. I would also suggest that you move the data presented in Figs S2 to S5 into the main section? You have no page or cost limitations by including them as a main observation instead of “hiding” them in your SI.
99. How were stats analyses performed for the Tab.1 data? The data in Tab. 1 A suggest that HAdV performs better in monoinfections. This could be interesting and should be discussed…
110. In line 200f, you write “However, coinfection with HAdV-5 also led to an increase in the severity of the clinical 200 manifestations.” Is this based on “subjective” observations? Could you provide any means to objectify and quantify these observations?
111. Fig. 1: The quality (resolution) is not sufficient. In this and other figures, it is not 100% clear which text belongs to the figure caption – that should be clarified! A and B labels are missing! I would be consistent with the abbreviation that are used for adenoviruses (HAdV-5 vs ad) as well as for SARS-CoV-2 (SARS-CoV-2 vs CoV)…
112. Fig. 2: A, B, C and D labels are missing!
113. The caption to Fig. 4 is incomplete (or does the paragraph starting with “Each of the six survey images…” belong to the figure?! Moreover, you write in lines 339f “Epithelial cell degeneration, vasculitis, and perivascular lymphocyte infiltration were observed in the lungs of IAV-infected animals.” Yet, you do not show these pathologies as pictures. These should be provided.
114. The pathological lesions should be scored by a pathologist to provide quantitative readouts (Figs 3 and 4) – see e.g. doi: 10.1165/rcmb.2020-0280LE as a reference. In addition, you should explain why the ferret lungs were not used for histopathological examinations in your study!
115. Please include detailed information about the stats tests in all your figure legends. Please also place the lines that you use to indicate statistical differences more accurately!
116. Paragraph 3.4. “Chemokine/cytokine responses in coinfected animals” is very short and superficial… as mentioned, these data could be included in the main manuscript and receive some more attention/descriptions.
117. Why were the chemokine/cytokine responses only measured in hamsters? The data would be much stronger if these were also performed with the ferret samples. Histopathology is also missing for the ferret samples…
118. It is very important to include more thorough comparisons of your data to previously published data on SARS-CoV-2 and IAV coinfections in animal models – like doi: 10.1038/s41598-021-00809-2, doi: 10.1128/jvi.01873-21, and doi: 10.1128/jvi.01791-21 for example.
119. Perform stats analyses on data presented in Figs S2 to S5
220. Scale bars are missing in Fig. S1.
221. Some parts of the SI are highlighted in yellow – did you do that on purpose?
222. Something’s odd in the legend to Fig. S4… Please double check “Body weight changes in monoinfected and coinfected with IAV and SARS-CoV-2 ham-Scheme 0”!
223. Fig. S5: Are the presented values the means ± SD of three independent experiments or of three different animals?
224. Author contributions: As per the International Committee of Medical Journal Editors, authorship has to be based on more than just funding acquisition or manuscript editing/proofreading, see https://www.icmje.org/recommendations/browse/roles-and-responsibilities/defining-the-role-of-authors-and-contributors.html. I recommend to discuss this issue with all authors and come up with an acceptable solution as I am unsure if the authors stick to the journal guidelines regarding their contributions to the paper and co-authorship. (PS: Alexander B. Ryzhikov is not even listed in the author contribution statement…).
Author Response
Thank you for considering our manuscript for publication in Microorganisms for special issue "Advances in SARS-CoV-2 Infection" titled “SARS-CoV-2 infection enhances by human adenovirus or influenza A virus in animal models”
We thank you for valuable suggestions that allowed us to make the manuscript more convincing and understandable. We accepted your suggestion and made corresponding changes in the manuscript.
Below please find our detailed responses to your questions and comments. All changes into MS (figures) are marked by in yellow (See attachment).
Yours sincerely,
Merry Christmas and Happy New Years!!!
Prof. (Dr.) Valery B. Loktev
You wrote:
Thanks for letting me review this draft. Here are some suggestions (major and minor) that should be addressed during preparation of a revised manuscript.
- Alternative title (suggestion): Human Adenovirus and Influenza A Virus Exacerbate SARS-CoV-2 Infection in Animal Models.
It is a good idea! The title of MS has been modified.
- Line 7: Is “World-Class Genomic Research Center” really part of your official affiliation?
Non official, only recommended by supervise of the Project’s curators. The affiliation has been modified to the official.
- In the very short introduction paragraph, include more information on animal models of the viruses used, see doi: 10.1016/j.coviro.2021.03.009 (SARS-CoV-2), doi: 10.3390/biology10121253 (HAdV), and doi: 10.1002/2211-5463.13416 (IAV) for example. Focus on significant information on hamsters and ferrets!
Small animal models are essential tools for viral pathogenesis, transmission, immunology and coinfection investigations [11-15]. The golden Syrian hamster and ferret models were usually used for SARS-CoV-2 and influenza infections. In SARS-CoV-2 challenge experiments, inoculated hamsters showed progressive weight loss with lethargy, ruffled fur, hunched back posture, and rapid breathing, with recovery by 14 days after virus inoculation [11]. The virus replicates to high titer in the upper and lower respiratory tracts, and the 50% infectious dose in Syrian hamsters is only five TCID50 [12]. SARS-CoV-2 causes pathological lung lesions including pulmonary edema and consolidation with evidence of interstitial pneumonia [12]. The Syrian hamster model also is the most commonly used human adenovirus animal model. Syrian hamsters that were used in HAdV-5 pathogenesis studies were found to have viral titers in blood and organ samples, and serologic as well as histologic evidence of infection [13]. Ferrets are commonly used in studies of influenza virus pathogenesis and transmission due to similarities to humans in receptor distribution and clinical course of the disease [14]. Ferrets are also susceptible to SARS-CoV-2 and the infected ferrets are showed mild clinical signs including elevated body temperature and reduced activity but no detectable body weight loss [15]. Viral shedding was mainly observed in the upper respiratory tracts but infectious viral titers were relatively low (1.83–2.88 log10 TCID50/mL) [15].
- In your M&M section, please indicate how many animals you used in total and how they were housed. Also, you mention the infection procedure incl. medicals used, but not how these animals were euthanized. That should be included.
The MS has been modified by these details.
- Line 85 and elsewhere: use superscript numbers (105 instead of 105 TCID50)
The MS has been modified.
- Were the nasopharyngeal wash samples collected under general anesthesia! Please clarify and include this information in the M&Ms if appropriate.
Infected animals were anaesthetized with isoflurane, nasal washes in ferrets and hamsters were obtained by administering 500µl and 100µl PBS to each nostril, respectively. The MS has been modified by these details. The MS has been modified.
- Line 137f: You used dPCR to compare genome loads between HAdV and SARS-CoV-2… a DNA virus vs. an RNA virus. For both, it seems you used cDNA as PCR sample, correct (see line 140)? Please clarify and revise.
The MS has been revised by next. Digital PCR (dPCR) was also used for the determination of the SARS-CoV-2 and HAdV-5 viral genome loads using the following primers (Table S1) [14]. The reaction mixture contained ddPCRSupermix (x2) (Bio–Rad), primers (900 nM), a probe (250 nM), cDNA or DNA.
- I would also suggest that you move the data presented in Figs S2 to S5 into the main section? You have no page or cost limitations by including them as a main observation instead of “hiding” them in your SI.
- The MS has been modified.
- How were stats analyses performed for the Tab.1 data? The data in Tab. 1 A suggest that HAdV performs better in monoinfections. This could be interesting and should be discussed…
Values represent means ± SD of three independent experiments. Lower HAdV-5 titer in coinfected cells at 48 and 72 hours postinfection due to SARS-CoV-2 induced lysis.
The MS has been modified.
- In line 200f, you write “However, coinfection with HAdV-5 also led to an increase in the severity of the clinical 200 manifestations.” Is this based on “subjective” observations? Could you provide any means to objectify and quantify these observations?
The coinfection of Syrian hamsters with SARS-CoV-2 and HAdV-5 and monoinfection with SARS-CoV-2 resulted in clinical signs such as: lethargy, ruffled fur, hunched back posture, and rapid breathing. Subjectively, these manifestations were slightly more pronounced in the coinfected hamsters (the clinical scores [36] for “CoV/ad”, “ad/3-days/CoV” and “CoV” groups were 3.60±1.15, 3.85±1.30 and 2.70±0.80, respectively).
The MS has been modified.
- Fig. 1: The quality (resolution) is not sufficient. In this and other figures, it is not 100% clear which text belongs to the figure caption – that should be clarified! A and B labels are missing! I would be consistent with the abbreviation that are used for adenoviruses (HAdV-5 vs ad) as well as for SARS-CoV-2 (SARS-CoV-2 vs CoV)…
The figure 1 has been modified and clarified.
- Fig. 2: A, B, C and D labels are missing!
The figures has been modified.
- The caption to Fig. 4 is incomplete (or does the paragraph starting with “Each of the six survey images…” belong to the figure?! Moreover, you write in lines 339f “Epithelial cell degeneration, vasculitis, and perivascular lymphocyte infiltration were observed in the lungs of IAV-infected animals.” Yet, you do not show these pathologies as pictures. These should be provided.
Formatting error. This is not a paragraph of text, but part of the legend to Figure 4. Fig.3 was also modified by additional signs: Epithelial cell degeneration (L), vasculitis (M), and perivascular lymphocyte infiltration (N) were observed in the lungs of IAV-infected animals
The MS and figures have been modified and clarified.
- The pathological lesions should be scored by a pathologist to provide quantitative readouts (Figs 3 and 4) – see e.g. doi: 10.1165/rcmb.2020-0280LE as a reference. In addition, you should explain why the ferret lungs were not used for histopathological examinations in your study!
The table with initial data for assessing pathological changes on a scale of 0-1-2-3 was added to additional materials. Scale: 0 - absence of pathology; 1- mild lesions; 2 - moderately lesions; 3 – pronounced lesions. The figure 3-4 were also modified as you recommended (ferret lungs and so on)
- Please include detailed information about the stats tests in all your figure legends. Please also place the lines that you use to indicate statistical differences more accurately!
The figure legends were modified as you recommended
- Paragraph 3.4. “Chemokine/cytokine responses in coinfected animals” is very short and superficial… as mentioned, these data could be included in the main manuscript and receive some more attention/descriptions.
The paragraph was modified as you recommended
- Why were the chemokine/cytokine responses only measured in hamsters? The data would be much stronger if these were also performed with the ferret samples. Histopathology is also missing for the ferret samples…
The additional information for histopathology was added as you recommended
- It is very important to include more thorough comparisons of your data to previously published data on SARS-CoV-2 and IAV coinfections in animal models – like doi: 10.1038/s41598-021-00809-2, doi: 10.1128/jvi.01873-21, and doi: 10.1128/jvi.01791-21 for example.
The additional references with appropriate discussion were added as you recommended
- Perform stats analyses on data presented in Figs S2 to S5
The MS has been modified by including this Figures to main part of MS
- Scale bars are missing in Fig. S1.
The MS (all figures) has been modified
- Some parts of the SI are highlighted in yellow – did you do that on purpose?
In yellow – it is marked changing to MS
- Something’s odd in the legend to Fig. S4… Please double check “Body weight changes in monoinfected and coinfected with IAV and SARS-CoV-2 ham-Scheme 0”!
The practically all figures has been modified
- Fig. S5: Are the presented values the means ± SD of three independent experiments or of three different animals?
The figure has been modified
- Author contributions: As per the International Committee of Medical Journal Editors, authorship has to be based on more than just funding acquisition or manuscript editing/proofreading, see https://www.icmje.org/recommendations/browse/roles-and-responsibilities/defining-the-role-of-authors-and-contributors.html. I recommend to discuss this issue with all authors and come up with an acceptable solution as I am unsure if the authors stick to the journal guidelines regarding their contributions to the paper and co-authorship. (PS: Alexander B. Ryzhikov is not even listed in the author contribution statement…).
The MS has been modified

Round 2
Reviewer 2 Report
It remains unclear why the chemokine/cytokine responses were only measured in hamsters. The paragraph title says “3.4. Chemokine/cytokine responses in coinfected animals” The data would be much stronger if these were also performed with the ferret samples. Histopathology is also missing for the ferret samples. The previous answer to this request was “The additional information for histopathology was added as you recommended” – yet, this has to be clarified further.
The authors also wrote “the figure 3-4 were also modified as you recommended (ferret lungs and so on)” in their rebuttal letter. This adds on to the abovementioned issue that is not yet sufficiently addressed in this manuscript.
Author Response
Comments for Reviewer 2
Many thanks you for considering our manuscript for publication in Microorganisms for special issue "Human Adenovirus and Influenza A Virus Exacerbate SARS-CoV-2 Infection in Animal Models ”
We thank you for valuable suggestions that allowed us to make the manuscript more convincing and understandable. We accepted your suggestion and made corresponding changes in the manuscript.
Below please find our detailed responses to your comments. The corresponding changes into MS (figures) are marked by in yellow.
Yours sincerely,
Prof. (Dr.) Valery B. Loktev
You wrote:
Comments and Suggestions for Authors
It remains unclear why the chemokine/cytokine responses were only measured in hamsters. The paragraph title says “3.4. Chemokine/cytokine responses in coinfected animals” The data would be much stronger if these were also performed with the ferret samples. Histopathology is also missing for the ferret samples. The previous answer to this request was “The additional information for histopathology was added as you recommended” – yet, this has to be clarified further.
You absolutely right that more detailed studying of the ferret model may be useful. But we afraid that ferret model was no well-studied for all detail experiments and preliminary data was shown that coronavirus infection was associated with local viral replication to upper respiratory tract of the ferret. Because for a more detailed study in coinfected SARS-CoV-2 with AdV-5 or with IAV animals (including histopathology and chemokine/cytokine studies) we used the most adequate and well-studied model to SARS-CoV-2 from considered (hamster model). The SARS-CoV-2 replicates to high titer in the upper and lower respiratory tracts, and the 50% infectious dose in Syrian hamsters is only five TCID50 (Imai M. et al., 2020, [MS ref. -12]). SARS-CoV-2 causes pathological lung lesions including pulmonary edema and consolidation with evidence of interstitial pneumonia (Imai M. et al., 2020, [12]). The Syrian hamster model also is the most commonly used human adenovirus animal model (Tollefson A. et al. 2017, [13] ).
Concerning the ferrets, no infectious SARS-CoV-2 was detected in the ferret lungs regardless of whether they were singly infected with SARS-CoV-2 or coinfected with IAV (Huang Y. et al., 2022, [36]), indicating that SARS-CoV-2 replicated in the upper but not the lower respiratory tract in ferrets. Infectious viral titers were low (1.83–2.88 lg TCID50/mL) (Kim Y. et al., 2020, [15]). It we additionally discussed in MS (see MS ref. above). It is marked by in yellow
We agree that an in-depth study of coinfection in ferrets is of interest and could be the subject of further research. We hope that additional information for ferret model will be obtained in future during more detailed investigation ferret as a model for IAV for accumulation all parameters for health for these animals.
The authors also wrote “the figure 3-4 were also modified as you recommended (ferret lungs and so on)” in their rebuttal letter. This adds on to the abovementioned issue that is not yet sufficiently addressed in this manuscript.
Histopathology is also missing for the ferret samples. The previous answer to this request was “The additional information for histopathology was added as you recommended” – yet, this has to be clarified further.
Also as we mention above that concerning the ferrets, no infectious SARS-CoV-2 (or very low titer) was detected in the ferret lungs regardless of whether they were singly infected with SARS-CoV-2 or coinfected with IAV (Huang Y. et al., 2022, [36]), indicating that SARS-CoV-2 replicated in the upper but not the lower respiratory tract in ferrets. Infectious viral titers were relatively low (1.83–2.88 lg TCID50/mL) (Kim Y. et al., 2020, [15]). It we additionally discussed in MS (reference 15, 36, 37) Marked by in yellow
Also, histopathological alterations in hamster lungs monoinfected with IAV were added in Fig.5 (former Fig. 3), Fig.5 L, M and N. Fig.6 includes the overview images of hamster (coinfected with IAV) lungs demonstrating that in double-infected animals with SARS-CoV-2 and IAV, zones of dense consolidation, oedema and cellular infiltration were generally more pronounced compared to monoinfection with IAV or SARS-CoV-2.
